# Exploring the impact of Brownian motion on novel closed-form solutions of the extended Kairat-II equation

Khaled Aldwoah[1], Alaa Mustafa[2], Tariq Aljaaidi [3]*, Khidir Mohamed[4], Amer Alsulami[5], Mohammed Hassan[6]

**1** Department of Mathematics, Faculty of Science, Islamic University of Madinah, Madinah, Saudi Arabia, **2** Department of Mathematics, Faculty of Science, Northern Border University, Arar, Saudi Arabia, **3** Department of Artificial Intelligence, Faculty of Computer Science and Information Technology, Alrazi University, Sana'a, Yemen, **4** Department of Mathematics, College of Science, Qassim University, Buraydah, Saudi Arabia, **5** Department of Mathematics, Turabah University College, Taif University, Taif, Saudi Arabia, **6** Department of Mathematics, Faculty of Science, University of Tabuk, Tabuk, Saudi Arabia

\* tariq10011@gmail.com

**Data Availability Statement:** All relevant data are within the manuscript and its Supporting Information files.

## Abstract

This work considers a stochastic form of an extended version of the Kairat-II equation by adding Browning motion into the deterministic equation. Two analytical approaches are utilized to derive analytical solutions of the modified equation. The first method is the modified Tanh technique linked with the Riccati equation, which is implemented to extract some closed-form solutions in the form of tangent and cotangent functions. The second technique is the Sardar sub-equation method (SSEM) which is used to attain several analytical solutions in the form of trigonometric and hyperbolic functions. Solutions selected randomly from the large families of solutions with suggested techniques are visualized in 3D and 2D scenarios. From the simulations an intriguing observation is made: the solutions generated through the modified tanh method exhibit a singular nature, with some of hybrid waves among them. On contrary to this, solutions derived through the SSEM, tend to be mostly non-singular in nature. The varying influence of the noise intensity revealed that the high amplitude and high energy regions of the waves are more vulnerable to the induced noise as compared to lower energy regions, which are relatively robust. This study introduces novel approaches by incorporating Brownian motion into the extended Kairat-II equation, providing new insights into the behavior of stochastic integrable systems that have not been previously explored.

## 1 Introduction

Integrable systems have garnered significant attention from researchers due to their numerous applications across various fields of science and engineering, such as fluid mechanics [1], quantum mechanics [2], optics [3], ocean engineering [4], and many others [5, 6]. The Kairat equations, as integrable systems, find utility in several areas of physical science, including

**Funding:** The author(s) received no specific funding for this work.

**Competing interests:** The authors have declared that no competing interests exist.

plasma, optical communication, and marine environments [7–10]. This paper focuses on the Kairat-II equation, which is expressed as:

$$\mathbb{G}_{xt} + \mathbb{G}_{xxxt} - 2\mathbb{G}_t\mathbb{G}_{xx} - 4\mathbb{G}_x\mathbb{G}_{xt} = 0. \tag{1}$$

Recently, Wazwaz extended the Kairat-II equation by adding three linear terms [11], resulting in the extended Kairat-II equation:

$$\mathbb{G}_{xt} + \mathbb{G}_{xxxt} - 2\mathbb{G}_t\mathbb{G}_{xx} - 4\mathbb{G}_x\mathbb{G}_{xt} + \alpha_1\mathbb{G}_{xx} + \alpha_2\mathbb{G}_{xy} + \alpha_3\mathbb{G}_{xz} = 0, \tag{2}$$

where $\mathbb{G} = \mathbb{G}(x, y, z, t)$ represents the wave profile, and $\alpha_1, \alpha_2, \alpha_3$ are parameters dependent on the specific context of the physical system.

Solitons are among the most stable solutions for integrable systems and have significant real-world applications. Researchers have developed various methods to derive different families of soliton solutions for integrable systems, such as the F-expansion method [12], the projective Riccati method [13], the exponential rational function method [14], the auxiliary equation method [15], and others [16, 17]. The modified Tanh method, linked with the Riccati equation, has been used to extract wave solutions in the form of tangent and cotangent functions for several equations, including the concatenation model [18], the Kundu–Mukherjee–Naskar equation [19], and others [20, 21]. Gaussian traveling wave solution of Schrödinger model is presented in [22]. Furthermore, SSEM has been employed to derive many analytical solutions in the form of trigonometric and hyperbolic functions, applied to various integrable systems [23, 24]. In similar way some more important solutions can be observed in [25–27].

A stochastic process, or random process, is a mathematical model that describes the probabilistic progression of a system or event over time. It involves a collection of random variables that evolve according to specific dynamics, resulting in uncertain outcomes that follow probabilistic patterns. Stochastic processes combined with differential equations are termed stochastic differential equations (SDEs), which have numerous applications in analyzing different physical systems [28–30]. In mathematical physics, particularly in soliton theory, stochastic processes can be incorporated into integrable systems by adding a noise term or Brownian motion. Various integrable systems with noise terms have been studied, including the Schrödinger equation [31], the SIdV equation [32], and the Chen–Lee–Liu equation [33]. Inspired by above works, we consider the Eq (2) in stochastic form as:

$$\mathbb{G}_{xt} + \mathbb{G}_{xxxt} - 2\mathbb{G}_t\mathbb{G}_{xx} - 4\mathbb{G}_x\mathbb{G}_{xt} + \alpha_1\mathbb{G}_{xx} + \alpha_2\mathbb{G}_{xy} + \alpha_3\mathbb{G}_{xz} + \vartheta(\mathbb{G}_x + \mathbb{G}_{xxx} - 2\mathbb{G}_x - 4\mathbb{G}_x\mathbb{G}_x)\frac{dW(t)}{dt} = 0, \tag{3}$$

where $\vartheta$ represents the strength of the noise, and $W(t)$ denotes the noise term, used for Brownian motion.

This study is the first as far as we know, to incorporate Brownian motion into the extended Kairat-II equation, providing new analytical solutions and insights into the impact of noise on soliton behaviors in various physical systems. Further, the comparison of the methods are also presented that which method is best for singular and which one for nonsingular soliton solutions. Besides this, deep interpretations of the varying noise is also not presented in previous studies.

For instance, the results obtained in work could have significant implications in fields like plasma physics, ocean engineering, and marine engineering, where understanding the effects of the noise on wave dynamics is crucial. The abbreviations and terms used throughout this study are presented in Table 1.

**Table 1. Acronyms and their descriptions.**

| Acronym | Description |
|---|---|
| SSEM | Sardar sub-equation method |
| SDEs | Stochastic Differential Equations |
| SIdV | Scale-invariant analogue of the Korteweg–de Vries equation |
| ODE | Ordinary Differential Equation |
| PDE | Partial Differential Equation |
| 2D | Two dimensional |
| 3D | Three dimensional |
| MRLW | Modified Regularized Long Wave |
| RLW | Regularized Long Wave |
| $\alpha_1, \alpha_2, \alpha_3$ | model parameters |
| $\gamma_1, \gamma_2, \gamma_3$ | Wave number |
| $\gamma_4$ | Wave speed |
| $\vartheta$ | Noise intensity |
| $W(t)$ | Stochastic noise |

## 2 Overview of the modified tanh method

This section outlines the algorithm and the steps which need to be followed for solving a non-linear partial differential equation (PDE) via the proposed technique. Consider the following general nonlinear PDE as:

$$\mathrm{L}(\delta, \delta_x, \delta_t, \delta_{xx}, \delta_{xt}, \ldots) = 0, \tag{4}$$

in this case $\delta = \delta(x, y, z, t)$. The suggested approach involves the following steps:

**Step I:** In the very first step, assume the solution in the following general form:

$$\delta(x, y, z, t) = \mathcal{F}(\psi), \tag{5}$$

in Eq (5), $\psi = \gamma_1 x + \gamma_2 y + \gamma_3 z - \gamma_4 t$. By inserting Eq (5) into Eq (4), the PDE can be transformed into an ODE with the following general form

$$\mathfrak{L}(\delta, \delta', \delta'', \delta''', \ldots) = 0, \tag{6}$$

here $\delta' = d\delta/d\psi$, $\delta'' = d^2\delta/d\psi^2$, $\delta''' = d^3\delta/d\psi^3$, ....

**Step II:** Here, assume the following expansion:

$$\mathcal{F}(\psi) = \mathrm{K}_0 + \sum_{\varpi=1}^{\rho} \mathrm{K}_\varpi \mathrm{L}^\varpi(\psi) + \mathrm{P}_\varpi \mathrm{L}^{-\varpi}(\psi), \ \ \mathrm{K}_\rho, \mathrm{P}_\rho \neq 0. \tag{7}$$

In Eq (7), the real parameters $\mathrm{K}_0, \mathrm{K}_1, \ldots, \mathrm{K}_\rho$ and $\mathrm{P}_1, \mathrm{P}_2, \ldots, \mathrm{P}_\rho$ are not yet specified and will be found later. Additionally, using the balance principle rule helps in finding the value of $\rho$. Additionally the function $\mathcal{F} = \mathcal{F}(\psi)$, represents the solution of ODE the Riccati equation:

$$\frac{d\mathrm{L}(\psi)}{d\psi} = \rho + \mathrm{L}(\psi)^2, \tag{8}$$

here $\rho$ is a real number, and the various general solutions based on the polarity of $\rho$ associated

to the above equation are:

$\rho < 0$

$$L_1(\psi) = -\sqrt{-\rho} \tanh(\mathcal{N}_1)$$

$$L_2(\psi) = -\sqrt{-\rho} \coth(\mathcal{N}_1)$$

$$L_3(\psi) = -\sqrt{-\rho}(\ \tanh(2\mathcal{N}_1)$$

$$+\imath\eta \sec h(2\mathcal{N}_1))$$

$$L_4(\psi) = \frac{\rho - \sqrt{\rho}\tanh(\mathcal{N}_1)}{1 + \sqrt{\rho}\tanh(\mathcal{N}_1)}$$

$$L_5(\psi) = \frac{-\sqrt{\rho}(5 - 4\cosh(2\mathcal{N}_1))}{3 + 4\sinh(2\mathcal{N}_1)}$$

$$L_6(\psi) = \frac{\eta\sqrt{-\rho(\varsigma^2 + \Upsilon^2)} - \varsigma\sqrt{-\rho}\cosh(2\mathcal{N}_1)}{\varsigma\sinh(2\mathcal{N}_1) + \Upsilon}$$

$$L_7(\psi) = \eta\sqrt{-\rho}\left(1 - \frac{2\varsigma}{\varsigma + \cosh(2\mathcal{N}_1) - \eta\sinh(2\mathcal{N}_1)}\right)$$

$$L_{15}(\psi) = -\frac{1}{\psi + \psi_0},\ \rho = 0$$

$\rho > 0$

$$L_8(\psi) = \sqrt{\rho}\tan(\mathcal{N}_2)$$

$$L_9(\psi) = -\sqrt{\rho}\cot(\mathcal{N}_2)$$

$$L_{10}(\psi) = \sqrt{\rho}(\ \tan(2\mathcal{N}_2)$$

$$+\imath\eta \sec(2\mathcal{N}_2))$$

$$L_{11}(\psi) = -\frac{\sqrt{\rho}(\ \tan(2\mathcal{N}_2))}{1 + \ \tan(\mathcal{N}_2)}$$

$$L_{12}(\psi) = \frac{\sqrt{\rho}(4 - 5\cos(2\mathcal{N}_2))}{3 + 5\sin(2\mathcal{N}_2)}$$

$$L_{13}(\psi) = \frac{\eta\sqrt{\rho(\varsigma^2 + \Upsilon^2)} - \varsigma\sqrt{\rho}\cos(2\mathcal{N}_2)}{\varsigma\sin(2\mathcal{N}_2) + \Upsilon}$$

$$L_{14}(\psi) = \imath\eta\sqrt{\rho}\left(1 - \frac{2\varsigma}{\varsigma + \cosh(2\mathcal{N}_2) - \imath\eta\sinh(2\mathcal{N}_2)}\right)$$

$\eta = \pm 1, \varsigma, \Upsilon, \rho, \psi_0\ are\ real\ constants.$

where $\mathcal{N}_1 = \sqrt{-\rho}(\psi + \psi_0)$ and $\mathcal{N}_2 = \sqrt{\rho}(\psi + \psi_0)$.

**Step III:** In the final step, putting Eq (7) into Eq (6) and equating the coefficients of various exponents of $L(\psi)$ to zero one can derive a system of algebraic equations. By solving the resultant system, the $K_0, K_1, \ldots, K_\rho$ and $P_1, P_2, \ldots, P_\rho$ can be obtained. Additionally, by plugging these values into Eq (6) and subsequently into Eq (5), a variety of solution families can be derived.

## 3 General procedure of the SSEM

In this portion, the steps for the proposed method are outlined. To do so, let us suppose the general form of the nonlinear PDE in (3+1) dimensions as follows:

$$\mathcal{C}(\mathcal{V}, \mathcal{V}_t, \mathcal{V}_x, \mathcal{V}_{tt}, \mathcal{V}_{xz}, \mathcal{V}_t, \mathcal{V}_{ty}, \cdots) = 0. \tag{9}$$

Next, let us assume the below transformation:

$$\mathcal{V}(x, y, z, t) = \beta(\psi). \tag{10}$$

In the transformation mentioned above, $\psi = \gamma_1 x + \gamma_2 y + \gamma_3 z - \gamma_4 t$. To solve Eq (9), the following steps are required:

**Step I:** In this step, it is necessary to insert Eq (10) into Eq (9) in order to convert the assumed PDE into ODE:

$$\mathcal{C}(\beta(\psi), \beta'(\psi), \beta''(\psi), \ldots) = 0. \tag{11}$$

In Eq (11), $\mathcal{C}$ is a polynomial in $\beta(\psi)$, where the superscripts (′) denote the ordinary derivatives of $\beta(\psi)$.

**Step II:** Here, in this step consider that the solution of Eq (11) is of the following form:

$$\beta(\psi) = \sum_{l=0}^{\kappa} \zeta_{\mathfrak{e}} \Phi(\psi)^l, \quad \zeta_{\mathfrak{e}} \neq 0, \tag{12}$$

where $\zeta_0, \zeta_1, \zeta_2, \ldots, \zeta_\kappa$ the values of the real constants to be determined, $\kappa$ belongs to positive integers. The value of $\kappa$ can be obtained using the homogeneous balance principle. Additionally $\Phi(\psi)$ satisfies the ODE:

$$\frac{d\Phi(\psi)}{d\psi} = \sqrt{\varpi_1 + \varpi_2 \Phi(\psi)^2 + \varpi_3 \Phi(\psi)^4}. \tag{13}$$

We have various types of solutions of Eq (13) such as:

**Case I:** When $\varpi_2 > 0$ and $\varpi_1 = 0$, then

$$\begin{aligned} \Phi_1^{\pm}(\psi) &= \pm\sqrt{-\mathrm{gh}\varpi_2} \, \sec h_{\mathrm{gh}}(\sqrt{\varpi_2}\psi) \\ \Phi_2^{\pm}(\psi) &= \pm\sqrt{\mathrm{gh}\varpi_2} \, \csc h_{\mathrm{gh}}(\sqrt{\varpi_2}\psi) \end{aligned}, \tag{14}$$

here

$$\sec h_{\mathrm{gh}}(\psi) = \frac{2}{\mathrm{g}e^\psi + \mathrm{h}e^{-\psi}}, \quad \csc h_{\mathrm{gh}}(\psi) = \frac{2}{\mathrm{g}e^\psi - \mathrm{h}e^{-\psi}}. \tag{15}$$

**Case II:** When $\varpi_2 < 0$ and $\varpi_1 = 0$, then

$$\begin{aligned} \Phi_3^{\pm}(\psi) &= \pm\sqrt{-\mathrm{gh}\varpi_2} \, \sec_{\mathrm{gh}}(\sqrt{-\varpi_2}\psi) \\ \Phi_4^{\pm}(\psi) &= \pm\sqrt{-\mathrm{gh}\varpi_2} \, \csc_{\mathrm{gh}}(\sqrt{-\varpi_2}\psi) \end{aligned}, \tag{16}$$

here

$$\sec_{\mathrm{gh}}(\psi) = \frac{2}{\mathrm{g}e^{\imath\psi} + \mathrm{h}e^{-\imath\psi}}, \quad \csc_{\mathrm{gh}}(\psi) = \frac{2\imath}{\mathrm{g}e^{\imath\psi} - \mathrm{h}e^{-\imath\psi}}. \tag{17}$$

**Case III:** When $\varpi_2 < 0$ and $\varpi_1 = \frac{\omega_2^2}{4}$, then

$$\Phi_5^{\pm}(\psi) = \pm\sqrt{-\frac{\varpi_2}{2}} \tanh_{\mathrm{gh}}\left(\sqrt{-\frac{\varpi_2}{2}}\psi\right)$$

$$\Phi_6^{\pm}(\psi) = \pm\sqrt{-\frac{\varpi_2}{2}} \coth_{\mathrm{gh}}\left(\sqrt{-\frac{\varpi_2}{2}}\psi\right)$$

$$\Phi_7^{\pm}(\psi) = \pm\sqrt{-\frac{\varpi_2}{2}}\left(\tanh_{\mathrm{gh}}(\sqrt{-2\varpi_2}\psi) + \imath\sqrt{-\mathrm{gh}}\,\sec h_{\mathrm{gh}}(\sqrt{-2\varpi_2}\psi)\right) \tag{18}$$

$$\Phi_8^{\pm}(\psi) = \pm\sqrt{-\frac{\varpi_2}{2}}\left(\coth_{\mathrm{gh}}(\sqrt{-2\varpi_2}\psi) + \sqrt{-\mathrm{gh}}\,\csc h_{\mathrm{gh}}(\sqrt{-2\varpi_2}\psi)\right)$$

$$\Phi_9^{\pm}(\psi) = \pm\sqrt{-\frac{\varpi_2}{2}}\left(\coth_{\mathrm{gh}}\left(\sqrt{-\frac{\varpi_2}{8}}\psi\right) + \imath\tanh_{\mathrm{gh}}\left(\sqrt{-\frac{\varpi_2}{8}}\psi\right)\right)$$

here

$$\tanh_{\mathrm{gh}}(\psi) = \frac{\mathrm{g}e^{\psi} - \mathrm{h}e^{-\psi}}{\mathrm{g}e^{\psi} + \mathrm{h}e^{-\psi}}, \ \ \coth_{\mathrm{gh}}(\psi) = \frac{\mathrm{g}e^{\psi} + \mathrm{h}e^{-\psi}}{\mathrm{g}e^{\psi} - \mathrm{h}e^{-\psi}}. \tag{19}$$

**Case IV:** When $\varpi_2 > 0$ and $\varpi_1 = \frac{\omega_2^2}{4}$, then

$$\Phi_{10}^{\pm}(\psi) = \pm\sqrt{\frac{\varpi_2}{2}} \tan_{\mathrm{gh}}\left(\sqrt{\frac{\varpi_2}{2}}\psi\right)$$

$$\Phi_{11}^{\pm}(\psi) = \pm\sqrt{\frac{\varpi_2}{2}} \cot_{\mathrm{gh}}\left(\sqrt{\frac{\varpi_2}{2}}\psi\right)$$

$$\Phi_{12}^{\pm}(\psi) = \pm\sqrt{\frac{\varpi_2}{2}}\left(\tan_{\mathrm{gh}}(\sqrt{2\varpi_2}\psi) + \sqrt{\mathrm{gh}} \sec_{\mathrm{gh}}(\sqrt{2\varpi_2}\psi)\right) \tag{20}$$

$$\Phi_{13}^{\pm}(\psi) = \pm\sqrt{\frac{\varpi_2}{2}}\left(\cot_{\mathrm{gh}}(\sqrt{2\varpi_2}\psi) + \sqrt{\mathrm{gh}} \csc_{\mathrm{gh}}(\sqrt{2\varpi_2}\psi)\right)$$

$$\Phi_{14}^{\pm}(\psi) = \pm\sqrt{\frac{\varpi_2}{2}}\left(\cot_{\mathrm{gh}}\left(\sqrt{\frac{\varpi_2}{8}}\psi\right) + \imath \tan_{\mathrm{gh}}\left(\sqrt{\frac{\varpi_2}{8}}\psi\right)\right)$$

here

$$\tan_{\mathrm{gh}}(\psi) = \frac{\mathrm{g}e^{\imath\psi} - \mathrm{h}e^{-\imath\psi}}{\mathrm{g}e^{\imath\psi} + \mathrm{h}e^{-\imath\psi}}, \ \ \cot_{\mathrm{gh}}(\psi) = \frac{\mathrm{g}e^{\imath\psi} + \mathrm{h}e^{-\imath\psi}}{\mathrm{g}e^{\imath\psi} - \mathrm{h}e^{-\imath\psi}}. \tag{21}$$

## 4 Applications

In this section, the methods described earlier are applied to derive and examine various new soliton solutions for the proposed Kairat-II equation. To do this, let's consider the following

$$\mathbb{G}(x, y, z, t) = \mathcal{F}(\psi) \, e^{9W(t) - 9t/2}, \ \psi = \gamma_1 x + \gamma_2 y + \gamma_3 z - \gamma_4 t. \tag{22}$$

Putting Eq (22) into Eq (3), one can get:

$$\gamma_1 \mathcal{F}''(\psi)(\alpha_1\gamma_1 + \alpha_2\gamma_2 + \alpha_3\gamma_3 + 6\gamma_1\gamma_4\mathcal{F}'(\psi) - \gamma_4) - \gamma_1^3\gamma_4\mathcal{F}^{(4)}(\psi) = 0, \tag{23}$$

by integrating Eq (23) with respect to $\psi$ and considering the constant of integration to be zero, we arrive at the following result

$$\gamma_1\left(\mathcal{F}'(\psi)(\alpha_1\gamma_1 + \alpha_2\gamma_2 + \alpha_3\gamma_3 - \gamma_4) + \gamma_1^2(-\gamma_4)\mathcal{F}^{(3)}(\psi)\right) + 3\gamma_1\gamma_4\mathcal{F}'(\psi)^2\Big) = 0. \tag{24}$$

### 4.1 Implementation of the modified tanh method

Through the homogeneous balance method, it is determined that $\rho = 2$ from Eq (24). Consequently, based on Eq (7), we get the following

$$\mathcal{F}(\psi) = \mathrm{K}_0 + \mathrm{K}_1\mathrm{L}^1(\psi) + \mathrm{P}_1\mathrm{L}^{-1}(\psi) + \mathrm{K}_2\mathrm{L}^2(\psi) + \mathrm{P}_2\mathrm{L}^{-2}(\psi), \ \mathrm{K}_1, \ \mathrm{P}_1 \neq 0. \tag{25}$$

By inserting Eq (25) into Eq (24) and incorporating Eq (8), the following result is derived
-1in0in

$$(\alpha_1\gamma_1 + \alpha_2\gamma_2 + \alpha_3\gamma_3 - \gamma_4)\left(K_1 L^1(\psi) + K_2 L(\psi)^2 + \frac{P_1}{L(\psi)} + \frac{P_2}{L(\psi)^2} + K_0\right)$$

$$+3\gamma_1\gamma_4\left(K_1 L(\psi) + K_2 L(\psi)^2 + \frac{P_1}{L(\psi))} + \frac{P_2}{L(\psi)^2} + K_0\right)^2$$

$$\gamma_1^2\gamma_4\left(2K_1 L(\psi)\left(L(\psi)^2 + \rho\right) + 4K_2 L(\psi)^2\left(L(\psi)^2 + \rho\right) + 2K_2\left(L(\psi)^2 + \rho\right)^2 + \frac{2P_1\left(L(\psi)^2 + \rho\right)^2}{L(\psi)^3}\right.$$

$$\left. - \frac{2P_1\left(L(\psi)^2 + \rho\right)}{L(\psi)} + \frac{6P_2\left(L(\psi)^2 + \rho\right)^2}{L(\psi)^4} - \frac{4P_2\left(L(\psi)^2 + \rho\right)}{L(\psi)^2}\right) = 0.$$

By comparing the coefficients for various powers of $L(\psi)$, the following is obtained

$$L^{-4}(\psi): \quad K_0(\alpha_1\gamma_1 + \alpha_2\gamma_2 + \alpha_3\gamma_3 - \gamma_4) + 3K_0^2\gamma_1\gamma_4 + 6K_1 P_1\gamma_1\gamma_4 - 2\rho^2 K_2\gamma_1^2\gamma_4 + 6K_2 P_2\gamma_1\gamma_4$$

$$-2P_2\gamma_1^2\gamma_4 = 0,$$

$$L^{-3}(\psi): \quad 3P_2^{\,2}\gamma_1\gamma_4 - 6\rho^2 P_2\gamma_1^2\gamma_4 = 0,$$

$$L^{-2}(\psi): \quad 6P_1 P_2\gamma_1\gamma_4 - 2\rho^2 P_1\gamma_1^2\gamma_4 = 0,$$

$$L^{-1}(\psi): \quad P_2(\alpha_1\gamma_1 + \alpha_2\gamma_2 + \alpha_3\gamma_3 - \gamma_4) + 6K_0 P_2\gamma_1\gamma_4 + 3P_1^{\,2}\gamma_1\gamma_4 - 8\rho P_2\gamma_1^2\gamma_4 = 0,$$

$$L^0(\psi): \quad P_1(\alpha_1\gamma_1 + \alpha_2\gamma_2 + \alpha_3\gamma_3 - \gamma_4) + 6K_0 P_1\gamma_1\gamma_4 + 6K_1 P_2\gamma_1\gamma_4 - 2\rho P_1\gamma_1^2\gamma_4 = 0, \qquad , (26)$$

$$L^1(\psi): \quad K_1(\alpha_1\gamma_1 + \alpha_2\gamma_2 + \alpha_3\gamma_3 - \gamma_4) + 6K_0 K_1\gamma_1\gamma_4 - 2\rho K_1\gamma_1^2\gamma_4 + 6K_2 P_1\gamma_1\gamma_4 = 0,$$

$$L^2(\psi): \quad K_2(\alpha_1\gamma_1 + \alpha_2\gamma_2 + \alpha_3\gamma_3 - \gamma_4) + 6K_0 K_2\gamma_1\gamma_4 + 3K_1^2\gamma_1\gamma_4 - 8\rho K_2\gamma_1^2\gamma_4 = 0,$$

$$L^3(\psi): \quad 6K_1 K_2\gamma_1\gamma_4 - 2K_1\gamma_1^2\gamma_4 = 0,$$

$$L^4(\psi): \quad 3K_2^2\gamma_1\gamma_4 - 6K_2\gamma_1^2\gamma_4 = 0.$$

Solving the system described above, we find the following values for the unknown parameters

$$\left\{K_0 = 4\rho\gamma_1, K_1 = 0, K_2 = 2\gamma_1, P_1 a = 0, P_2 = 2\rho^2\gamma_1, \gamma_4 = \frac{-\alpha_1\gamma_1 - \alpha_2\gamma_2 - \alpha_3\gamma_3}{16\rho\gamma_1^{\,2} - 1}\right.$$

Incorporating Eq (25) along with various solutions for $\mathcal{F}(\psi)$ into Eq (22), general solutions of the proposed equation can be derived. Each set of substitutions results in several families of

solutions. When $\rho < 0$, we obtained:

$$\mathbb{G}_1(x,y,z,t) = e^{9W(t)-9t/2}\left(K_0 - K_1\sqrt{-\rho}\tanh(\mathcal{N}_1) - \frac{P_1\coth(\mathcal{N}_1)}{\sqrt{-\rho}}\right)$$

$$\mathbb{G}_2(x,y,z,t) = e^{9W(t)-9t/2}\left(K_0 - K_1\sqrt{-\rho}\coth(\mathcal{N}_1) - \frac{P_1\tanh(\mathcal{N}_1)}{\sqrt{-\rho}}\right)$$

$$\mathbb{G}_3(x,y,z,t) = e^{9W(t)-9t/2}\left(K_0 + K_1\left(-\sqrt{-\rho}\tanh(2\mathcal{N}_1) + i\eta\,\mathrm{sech}(2\mathcal{N}_1)\right) + \frac{P_1}{-\sqrt{-\rho}\tanh(2\mathcal{N}_1) + i\eta\,\mathrm{sech}(2\mathcal{N}_1)}\right)$$

$$\mathbb{G}_4(x,y,z,t) = e^{9W(t)-9t/2}\left(K_0 + \frac{K_1(\rho - \sqrt{-\rho}\tanh(\mathcal{N}_1))}{\sqrt{-\rho}\tanh(\mathcal{N}_1) + 1} + \frac{P_1(\sqrt{-\rho}\tanh(\mathcal{N}_1) + 1)}{\rho - \sqrt{-\rho}\tanh(\mathcal{N}_1)}\right)$$

$$\mathbb{G}_5(x,y,z,t) = e^{9W(t)-9t/2}\left(K_0 + \frac{K_1\sqrt{-\rho}(5 - 4\cosh(2\sqrt{-\rho}(\psi0+\psi1)))}{4\sinh(2\mathcal{N}_1) + 3} + \frac{P_1(4\sinh(2\mathcal{N}_1) + 3)}{\sqrt{-\rho}(5 - 4\cosh(2\mathcal{N}_1))}\right)$$

$$\mathbb{G}_6(x,y,z,t) = e^{9W(t)-9t/2}\left(K_0 + \frac{K_1(\eta\sqrt{-(\rho(\varsigma^2 + \Upsilon^2))} - \varsigma\sqrt{-\rho}\cosh(2\mathcal{N}_1))}{\varsigma\sinh(2\mathcal{N}_1) + \Upsilon} + \frac{P_1(\varsigma\sinh(2\mathcal{N}_1) + \Upsilon)}{\eta\sqrt{-(\rho(\varsigma^2 + \Upsilon^2))} - \varsigma\sqrt{-\rho}\cosh(2\mathcal{N}_1)}\right)$$

$$\mathbb{G}_7(x,y,z,t) = e^{9W(t)-9t/2}\left(K_0 + \frac{P_1}{\eta\sqrt{-\rho}\left(1 - \frac{2\varsigma}{-\eta\sinh(2\mathcal{N}_1)+\varsigma+\cosh(2\mathcal{N}_1)}\right)} + K_1\eta\sqrt{-\rho}\left(1 - \frac{2\varsigma}{-\eta\sinh(2\mathcal{N}_1) + \varsigma + \cosh(2\mathcal{N}_1)}\right)\right)$$

when $\rho > 0$ the solution are as follows:

$$\mathbb{G}_8(x,y,z,t) = e^{9W(t)-9t/2}\left(K_0 - K_1\sqrt{\rho}\tan(\mathcal{N}_2) + \frac{P_1\cot(\mathcal{N}_2)}{\sqrt{\rho}}\right)$$

$$\mathbb{G}_9(x,y,z,t) = e^{9W(t)-9t/2}\left(K_0 - K_1\sqrt{\rho}\cot(\mathcal{N}_2) - \frac{P_1\tan(\mathcal{N}_2)}{\sqrt{-\rho}}\right)$$

$$\mathbb{G}_{10}(x,y,z,t) = e^{9W(t)-9t/2}\left(K_0 + K_1\left(\sqrt{\rho}\tan(2\mathcal{N}_2) + \eta\sec(2\mathcal{N}_2)\right) + \frac{P_1}{\sqrt{\rho}\tan(2\mathcal{N}_2) + \eta\sec(2\mathcal{N}_2)}\right)$$

$$\mathbb{G}_{11}(x,y,z,t) = e^{9W(t)-9t/2}\left(K_0 - \frac{K_1\sqrt{\rho}(1 - \tan(\mathcal{N}_2))}{\tan(\mathcal{N}_2) + 1} - \frac{P_1(\tan(\mathcal{N}_2) + 1)}{\sqrt{\rho}(1 - \tan(\mathcal{N}_2))}\right)$$

$$\mathbb{G}_{12}(x,y,z,t) = e^{9W(t)-9t/2}\left(K_0 + \frac{K_1\sqrt{\rho}(5 - 4\cos(2\mathcal{N}_2))}{4\sin(2\mathcal{N}_2) + 3} + \frac{P_1(4\sin(2\mathcal{N}_2) + 3)}{\sqrt{\rho}(5 - 4\cos(2\mathcal{N}_2))}\right)$$

$$\mathbb{G}_{13}(x,y,z,t) = e^{9W(t)-9t/2}\left(K_0 + \frac{K_1(\eta\sqrt{(\rho(\varsigma^2 + \Upsilon^2))} - \varsigma\sqrt{\rho}\cosh(2\mathcal{N}_2))}{\varsigma\sinh(2\mathcal{N}_2) + \Upsilon} + \frac{P_1(\varsigma\sinh(2\mathcal{N}_2) + \Upsilon)}{\eta\sqrt{(\rho(\varsigma^2 + \Upsilon^2))} - \varsigma\sqrt{\rho}\cosh(2\mathcal{N}_2)}\right)$$

$$\mathbb{G}_{14}(x,y,z,t) = e^{9W(t)-9t/2}\left(K_0 + K_1\eta\sqrt{\rho}\left(1 - \frac{2\varsigma}{-\eta\sin(2\mathcal{N}_2) + \varsigma + \cos(2\mathcal{N}_2)}\right) + \frac{P_1}{\eta\sqrt{\rho}\left(1 - \frac{2\varsigma}{-\eta\sin(2\mathcal{N}_2)+\varsigma+\cos(2\mathcal{N}_2)}\right)}\right)$$

## 4.2 Implementation of the SSEM

In the light of homogeneous balance technique, it is determined that $\kappa = 2$. Therefore, based on Eq (24), we can conclude that

$$\beta(\psi) = \zeta_0 + \zeta_1 \Phi(\psi) + \zeta_2 \Phi^2(\psi), \qquad \zeta_1 \ or \ \zeta_2 \ or \ \zeta_3 \neq 0, \tag{27}$$

where in Eq (27), the function $\Phi(\psi)$ satisfies the following ODE

$$\frac{d\Phi(\psi)}{d\psi} = \sqrt{\varpi_1 + \varpi_2 \Phi(\psi)^2 + \varpi_3 \Phi(\psi)^4}. \tag{28}$$

By substituting Eq (28) into Eq (27), and then utilizing Eq (24), and equating different powers of $\Phi(\psi)$. This leads to the following algebraic system

$$\Phi(\psi)^0 : \quad \alpha_1 \zeta_0 \gamma_1 + 3\zeta_0^2 \gamma_1 \gamma_4 - \zeta_0 \gamma_4 + \zeta_0 \alpha_2 \gamma_2 + \zeta_0 \alpha_3 \sigma 3 - 2\zeta_2 \varpi_1 \gamma_1^2 \gamma_4 = 0,$$

$$\Phi(\psi)^1 : \quad \alpha_1 \zeta_1 \gamma_1 + 6\zeta_0 \zeta_1 \gamma_1 \gamma_4 - \zeta_1 \gamma_4 + \zeta_1 \alpha_2 \gamma_2 + \zeta_1 \alpha_3 \sigma 3 + \zeta_1 (-\varpi_2) \gamma_1^2 \gamma_4 = 0,$$

$$\Phi(\psi)^2 : \quad \alpha_1 \zeta_2 \gamma_1 + 6\zeta_0 \zeta_2 \gamma_1 \gamma_4 + 3\zeta_1^2 \gamma_1 \gamma_4 - \zeta_2 \gamma_4 + \zeta_2 \alpha_2 \gamma_2 + \zeta_2 \alpha_3 \sigma 3 - 4\zeta_2 \varpi_2 \gamma_1^2 \gamma_4 = 0,$$

$$\Phi(\psi)^3 : \quad 6\zeta_1 \zeta_2 \gamma_1 \gamma_4 - 2\zeta_1 \varpi_3 \gamma_1^2 \gamma_4 = 0,$$

$$\Phi(\psi)^4 : \quad 3\zeta_2^2 \gamma_1 \gamma_4 - 6\zeta_2 \varpi_3 \gamma_1^2 \gamma_4 = 0.$$

Solving this system, we derive the following non-trivial parameter values

$$\alpha_1 = 0, \varpi_1 = \frac{\zeta_0(\alpha_1 \gamma_1 + 3\zeta_0 \gamma_1 \gamma_4 + \alpha_2 \gamma_2 + \alpha_3 \sigma 3 - \gamma_4)}{2\zeta_2 \gamma_1^2 \gamma_4}, \varpi_2 = \frac{\alpha_1 \gamma_1 + 6\zeta_0 \gamma_1 \gamma_4 + \alpha_2 \gamma_2 + \alpha_3 \gamma_3 - \gamma_4}{4\gamma_1^2 \gamma_4}, \varpi_3 = \frac{\zeta_2}{2\gamma_1}.$$

By plugging these parameters into Eqs (14), (16), (18) and (20), the following analytical solutions for the given equation

**Case I:** When $\varpi_2 > 0$ and $\varpi_1 = 0, \Rightarrow \zeta_0 = 0$, then

$$\mathbb{G}_{15}^{\pm}(x, y, z, t) = \pm \zeta_2 \sqrt{gh\varpi_2}\, e^{\vartheta W(t) - \vartheta t/2} \sec h_{gh}(\sqrt{\varpi_2}(\mathrm{x} - \mathrm{vt})), \tag{29}$$

the requirement for the existence of the above result is that gh < 0.

$$\mathbb{G}_{16}^{\pm}(x, y, z, t) = \pm \zeta_2 \sqrt{gh\varpi_2}\, e^{\vartheta W(t) - \vartheta t/2} \csc h_{gh}(\sqrt{\varpi_2}(\mathrm{x} - \mathrm{vt})), \tag{30}$$

the requirement for the existence of the above result is that gh < 0.

**Case II:** When $\varpi_2 < 0$ and $\varpi_1 = 0$, then

$$\mathbb{G}_{17}^{\pm}(x, y, z, t) = \pm \zeta_2 \sqrt{-gh\varpi_2}\, e^{\vartheta W(t) - \vartheta t/2} \sec_{gh}(\sqrt{-\varpi_2}(\mathrm{x} - \mathrm{vt})), \tag{31}$$

$$\mathbb{G}_{18}^{\pm}(x, y, z, t) = \pm \zeta_2 \sqrt{-gh\varpi_2}\, e^{\vartheta W(t) - \vartheta t/2} \csc_{gh}(\sqrt{-\varpi_2}(\mathrm{x} - \mathrm{vt})), \tag{32}$$

the condition for the existence of the above solution is that gh > 0.

**Case III:** When $\varpi_2 < 0$ and $\varpi_1 = \frac{\omega_2^2}{4}$, then

$$\mathbb{G}_{19}^{\pm}(x, y, z, t) = \pm\zeta_2\sqrt{-\frac{\varpi_2}{2}}e^{9W(t)-9t/2}\tanh_{\mathrm{gh}}\left(\sqrt{-\frac{\varpi_2}{2}}(\mathrm{x}-\mathrm{vt})\right)$$

$$\mathbb{G}_{20}^{\pm}(x, y, z, t) = \pm\zeta_2\sqrt{-\frac{\varpi_2}{2}}e^{9W(t)-9t/2}\coth_{\mathrm{gh}}\left(\sqrt{-\frac{\varpi_2}{2}}(\mathrm{x}-\mathrm{vt})\right)$$

$$\mathbb{G}_{21}^{\pm}(x, y, z, t) = \pm\zeta_2\sqrt{-\frac{\varpi_2}{2}}e^{9W(t)-9t/2}\left(\tanh_{\mathrm{gh}}\left(\sqrt{-2\varpi_2}(\mathrm{x}-\mathrm{vt})\right) + \iota\sqrt{\mathrm{gh}}\sec h_{\mathrm{gh}}\left(\sqrt{-2\varpi_2}(\mathrm{x}-\mathrm{vt})\right)\right).$$

$$\mathbb{G}_{22}^{\pm}(x, y, z, t) = \pm\zeta_2\sqrt{-\frac{\varpi_2}{2}}e^{9W(t)-9t/2}\left(\coth_{\mathrm{gh}}\left(\sqrt{-2\varpi_2}(\mathrm{x}-\mathrm{vt})\right) + \sqrt{\mathrm{gh}}\csc h_{\mathrm{gh}}\left(\sqrt{-2\varpi_2}(\mathrm{x}-\mathrm{vt})\right)\right)$$

$$\mathbb{G}_{23}^{\pm}(x, y, z, t) = \pm\zeta_2\sqrt{-\frac{\varpi_2}{2}}e^{9W(t)-9t/2}\left(\coth_{\mathrm{gh}}\left(\sqrt{-\frac{\varpi_2}{8}}(\mathrm{x}-\mathrm{vt})\right) + \iota\tanh_{\mathrm{gh}}\left(\sqrt{-\frac{\varpi_2}{8}}(\mathrm{x}-\mathrm{vt})\right)\right).$$

**Case IV:** When $\varpi_2 > 0$ and $\varpi_1 = \frac{\omega_2^2}{4}$, then

$$\mathbb{G}_{24}^{\pm}(x, y, z, t) = \pm\zeta_2\sqrt{\frac{\varpi_2}{2}}e^{9W(t)-9t/2}\tan_{\mathrm{gh}}\left(\sqrt{\frac{\varpi_2}{2}}(\mathrm{x}-\mathfrak{L}\mathrm{t})\right)$$

$$\mathbb{G}_{25}^{\pm}(x, y, z, t) = \pm\zeta_2\sqrt{\frac{\varpi_2}{2}}e^{9W(t)-9t/2}\cot_{\mathrm{gh}}\left(\sqrt{\frac{\varpi_2}{2}}(\mathrm{x}-\mathfrak{L}\mathrm{t})\right)$$

$$\mathbb{G}_{26}^{\pm}(x, y, z, t) = \pm\zeta_2\sqrt{\frac{\varpi_2}{2}}e^{9W(t)-9t/2}\left(\tan_{\mathrm{gh}}\left(\sqrt{2\varpi_2}(\mathrm{x}-\mathrm{vt})\right) + \sqrt{\mathrm{gh}}\sec_{\mathrm{gh}}\left(\sqrt{2\varpi_2}(\mathrm{x}-\mathrm{vt})\right)\right).$$

$$\mathbb{G}_{27}^{\pm}(x, y, z, t) = \pm\zeta_2\sqrt{\frac{\varpi_2}{2}}e^{9W(t)-9t/2}\left(\cot_{\mathrm{gh}}\left(\sqrt{2\varpi_2}(\mathrm{x}-\mathrm{vt})\right) + \sqrt{\mathrm{gh}}\csc_{\mathrm{gh}}\left(\sqrt{2\varpi_2}(\mathrm{x}-\mathrm{vt})\right)\right)$$

$$\mathbb{G}_{28}^{\pm}(x, y, z, t) = \pm\zeta_2\sqrt{\frac{\varpi_2}{2}}e^{9W(t)-9t/2}\left(\cot_{\mathrm{gh}}\left(\sqrt{\frac{\varpi_2}{8}}(\mathrm{x}-\mathrm{vt})\right) + \iota\tan_{\mathrm{gh}}\left(\sqrt{\frac{\varpi_2}{8}}(\mathrm{x}-\mathrm{vt})\right)\right).$$

## 5 Graphical investigation and discussions

In this section, we perform numerical simulations for a selection of the exact solutions of the Eq (3) obtained with the proposed two different methods. Each figure comprises six sub-plots: sub-plots (a), (b), and (c) illustrate the 3D behavior of the results, while sub-plots (d), (e), and (f) depict the 2D dynamics, taking into account various values of noise parameter.

In the figures presented in this section, sub-plots [(a), (d)] display the deterministic behavior without noise terms, while others represent the effects of noise parameters on the wave dynamics of exact solutions.

Fig 1 presents the physical behavior of the exact solution $\mathbb{G}_3$. For this simulation, the parameters are used in the form of $y = z = 1, \alpha_1 = \alpha_2 = \alpha_3 = 1, \gamma_1 = \gamma_2 = \gamma_3 = \gamma_4 = 1, \eta = \psi_0 = 1, \rho = -1$. In the simulation of the solution $\mathbb{G}_3$, we observed the emergence of a singular solitary wave. Remarkably, it became evident that the stochastic term profoundly influences the dynamics of the wave. Over time, we noticed a discernible effect: the stochastic term gradually diminishes the gap between the two sides of the wave's crest. This observation suggests a tendency towards a reduction in the spatial extent of the wave's peak as time progresses, indicative

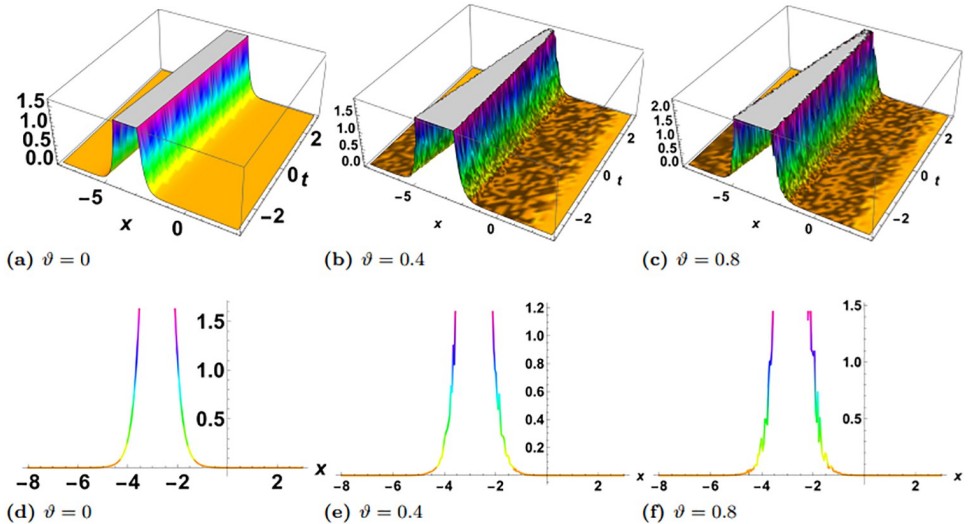

**Fig 1. Visual illustration of the solution $\mathbb{G}_3(x, y, z, t)$, taking into account specific parameters $y = z = 1, \alpha_1 = \alpha_2 = \alpha_3 = 1, \gamma_1 = \gamma_2 = \gamma_3 = \gamma_4 = 1, \eta = \psi_0 = 1, \rho = -1$.**

of an intriguing interplay between deterministic and stochastic components in shaping wave behavior. These dynamics are more clearly visulaized in Fig 2, where in left plot $\vartheta = 0$ and $t = -20, 0, 20$, in right plot $\vartheta = 0.2$ and $t = 2, 5, 10$. Clearly it can be observed that there is no loss in the amplitude when there is no noise, contrary to this the amplitude decreases with time when $\vartheta = 0.4$.

Fig 3 shows the physical dynamics of the solution $\mathbb{G}_5$ with the parameters' selection as $y = z = 1, \alpha_1 = \alpha_2 = \alpha_3 = 1, \gamma_1 = \gamma_2 = \gamma_3 = \gamma_4 = 1, \psi_0 = 1, \rho = -0.004$. The graphs show the dark singular solitary wave nature, where the noise term affects the wave separation and decrease it with time. Furthermore the simulation of the exact solution $\mathbb{G}_8$ is carried out with parameters $y = z = 1, \alpha_1 = \alpha_2 = \alpha_3 = 1, \gamma_1 = \gamma_2 = \gamma_3 = \gamma_4 = 1, \psi_0 = 1, \rho = 0.3$ in Fig 4. Here, we observed singular periodic waves. From the variation in noise term $\vartheta$, we see that the lower amplitude areas become more random as compared to high amplitudes.

Fig 5 demonstrates the exact solution $\mathbb{G}_{10}$, with the supposition of the values of parameters as $y = z = 1, \alpha_1 = \alpha_2 = \alpha_3 = 1, \gamma_1 = \gamma_2 = \gamma_3 = \gamma_4 = 1, \psi_0 = 1, \rho = 0.3, \eta = 1$. Here some hybrid wave including bright and dark singular waves is observed. In the same fashion, the exact

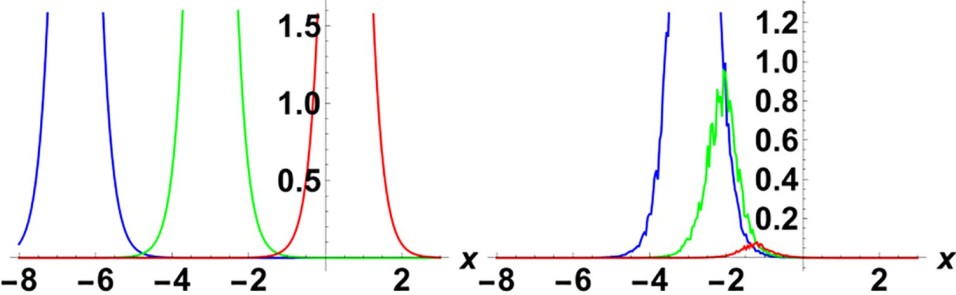

**Fig 2. Visualization of the $\mathbb{G}_3(x, y, z, t)$, taking into account specific parameters $y = z = 1, \alpha_1 = \alpha_2 = \alpha_3 = 1, \gamma_1 = \gamma_2 = \gamma_3 = \gamma_4 = 1, \eta = \psi_0 = 1, \rho = -1$, left plot $\vartheta = 0$ and right plot $\vartheta = 0.4$.**

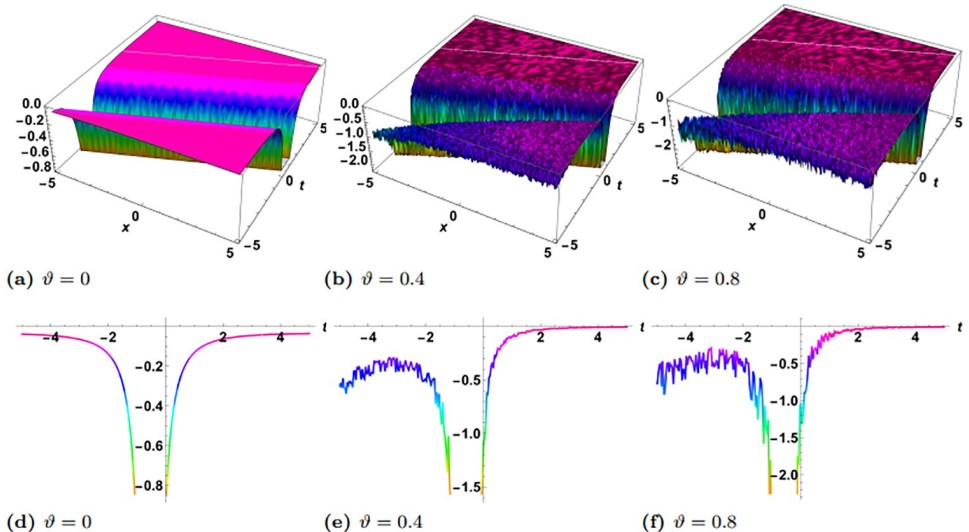

**Fig 3. Visual illustration of the solution** $\mathbb{G}_5(x, y, z, t)$, **taking into account specific parameters** $y = z = 1$, $\alpha_1 = \alpha_2 = \alpha_3 = 1$, $\gamma_1 = \gamma_2 = \gamma_3 = \gamma_4 = 1$, $\psi_0 = 1$, $\rho = -0.004$.

solution $\mathbb{G}_{12}$ is depicted in Fig 6 with $y = z = 1$, $\alpha_1 = \alpha_2 = \alpha_3 = 1$, $\gamma_1 = \gamma_2 = \gamma_3 = \gamma_4 = 1$, $\psi_0 = 1$, $\rho = 0.002$. Here also a hybrid singular wave is observed, where one can see that kink is emerging as the $\vartheta$ increases.

Some of the solutions obtained with the SSe approach are demonstrated in the Figs 7, 9 and 10. The Fig 7 shows the exact solution $\mathbb{G}_{15}(x, y, z, t)$ with the choice of parameters in the form of $y = z = 1$, $\alpha_1 = \alpha_2 = \alpha_3 = 1$, $\gamma_1 = \gamma_2 = \gamma_3 = \gamma_4 = 1$, $\xi_0 = 0$, $\xi_2 = 1$, $g = 1$, $h = 1$. From the simulations of the results bright solitary wave is observed where the increase in noise parameter affects the amplitude of the wave and decreases with time. The decrease in amplitude in the proposed solution where as time evolves is presented as 2D plot in Fig 8, where the time is

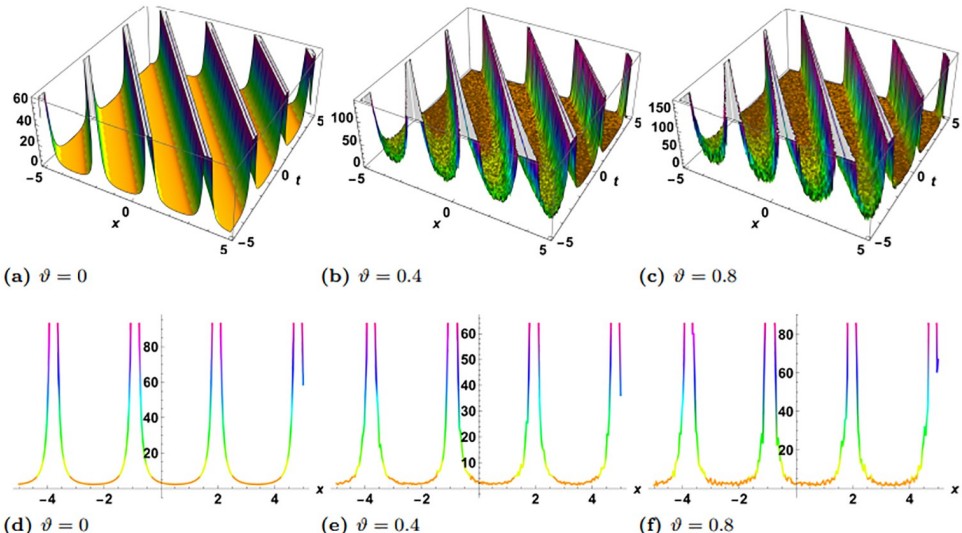

**Fig 4. Visual illustration of the solution** $\mathbb{G}_8(x, y, z, t)$, **taking into account specific parameters** $y = z = 1$, $\alpha_1 = \alpha_2 = \alpha_3 = 1$, $\gamma_1 = \gamma_2 = \gamma_3 = \gamma_4 = 1$, $\psi_0 = 1$, $\rho = 0.3$.

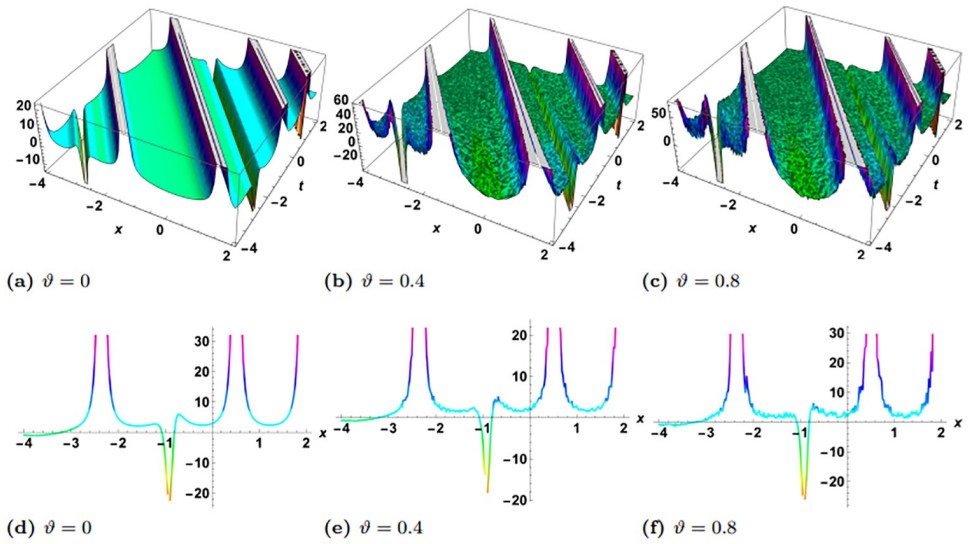

**Fig 5. Visual illustration of the solution** $\mathbb{G}_{10}(x, y, z, t)$**, taking into account specific parameters** $y = z = 1$, $\alpha_1 = \alpha_2 = \alpha_3 = 1$, $\gamma_1 = \gamma_2 = \gamma_3 = \gamma_4 = 1$, $\psi_0 = 1$, $\rho = 0.3$, $\eta = 1$.

considered as blue $t = -5$, green $t = 0$, and red $t = 5$. In the left plot no stochatic noise is induced, such that $\vartheta = 0$, while in right plot $\vartheta = 0.2$ From the figure it is clearly visible that as time evolves, due to stochastic effect the amplitude decreases, on contrary to this the solution with zero noise propagates with no damping. Furthermore, the exact solution $\mathbb{G}_{16}(x, y, z, t)$ is demonstrated in the Fig 9, with parameters $y = z = 1$, $\alpha_1 = \alpha_2 = \alpha_3 = 1$, $\gamma_1 = \gamma_2 = \gamma_3 = \gamma_4 = 1$, $\xi_0 = 0$, $\xi_2 = 1$, g = 1, h = 1. Here the periodic wave is observed. Our observations indicate that these fluctuations are more pronounced in areas where the wave reaches its highest and lowest points (the peaks and troughs), which correspond to higher amplitude regions. In contrast, the noise has a lesser effect in areas where the wave is closer to its average value, or lower

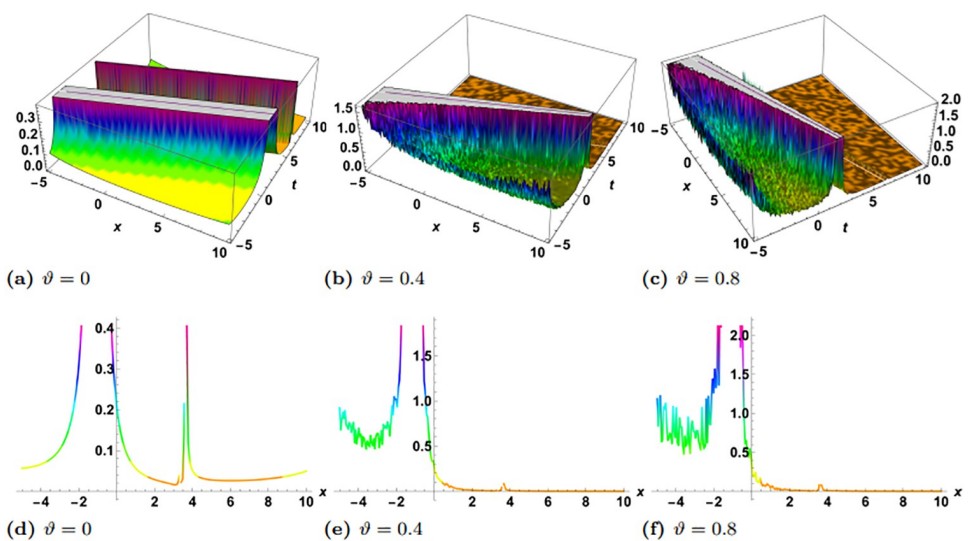

**Fig 6. Visual illustration of the solution** $\mathbb{G}_{12}(x, y, z, t)$**, taking into account specific parameters** $y = z = 1$, $\alpha_1 = \alpha_2 = \alpha_3 = 1$, $\gamma_1 = \gamma_2 = \gamma_3 = \gamma_4 = 1$, $\psi_0 = 1$, $\rho = 0.002$.

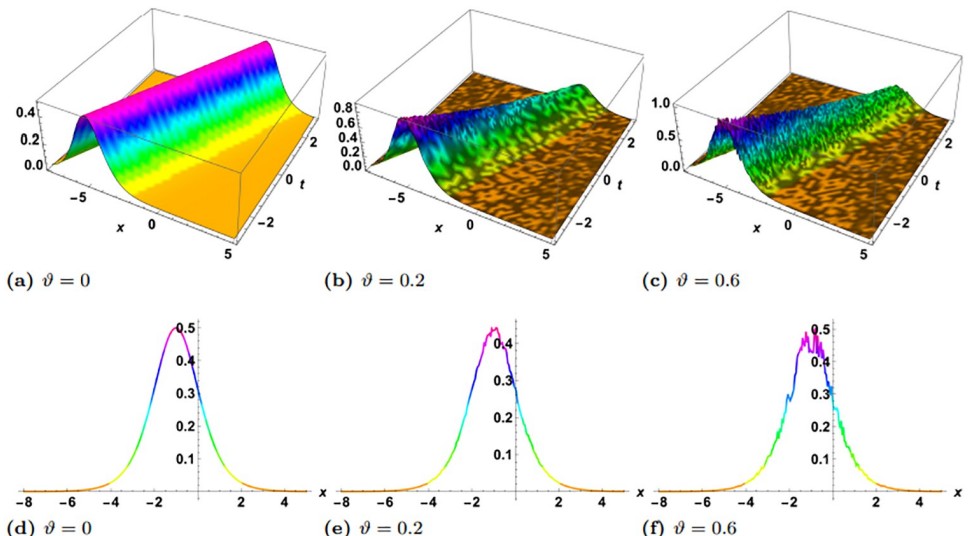

**Fig 7. Visual illustration of the solution** $\mathbb{G}_{15}(x, y, z, t)$, **taking into account specific parameters** $y = z = 1$, $\alpha_1 = 0.5$, $\alpha_2 = \alpha_3 = 1$, $\gamma_1 = \gamma_2 = \gamma_3 = \gamma_4 = 1$, $\xi_0 = 0$, $\xi_2 = 1$, $g = 1$, $h = 1$.

amplitude regions. This pronounced effect of noise on high-energy regions may be attributed to the inherent instability of these areas in response to perturbations. High-energy regions correspond to wave peaks where gradients are steep, making them more susceptible to fluctuations caused by stochastic noise. As noise is introduced, these regions absorb the variations more acutely due to the greater energy concentration, leading to amplified distortions. Additionally, in physical terms, high-energy points often exhibit increased sensitivity to external disturbances, as the energy stored in these areas can facilitate a stronger and faster response to stochastic forces. Finally the exact solution $\mathbb{G}_{17}(x, y, z, t)$ is demonstrated in the Fig 10, where the parameters are supposed as $y = z = 1$, $\alpha_1 = \alpha_2 = \alpha_3 = 1$, $\gamma_1 = \gamma_2 = \gamma_3 = \gamma_4 = -1$, $\xi_0 = 0$, $\xi_2 = 1$, $g = 1$, $h = 1$. The simulation shows the dark solitary wave behavior, with increase in the noise the randomness affect the whole wave.

In plasma systems, soliton solutions often represent stable energy wave packets. The influence of the stochastic noise in such solutions could model external disturbances in the plasma, like magnetic or electric field fluctuations. In marine environments, these solitonic forms

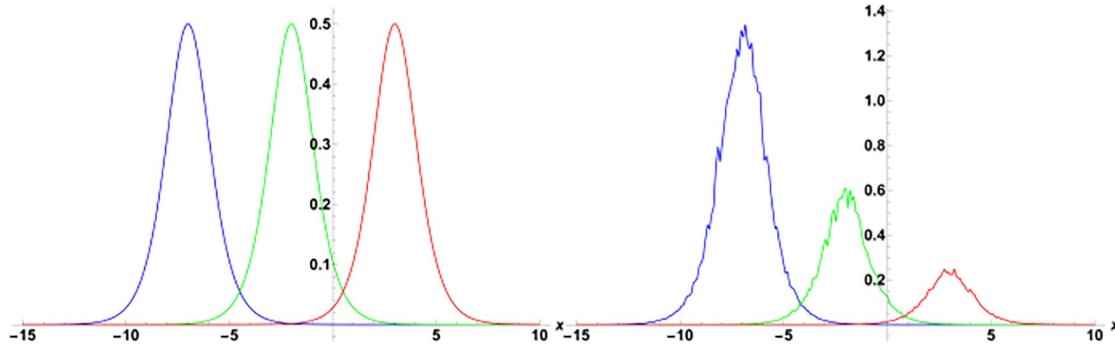

**Fig 8. Visualization of the solution** $\mathbb{G}_{15}(x, y, z, t)$, **taking into account specific parameters** $y = z = 1$, $\alpha_1 = 0.5$, $\alpha_2 = \alpha_3 = 1$, $\gamma_1 = \gamma_2 = \gamma_3 = \gamma_4 = 1$, $\xi_0 = 0$, $\xi_2 = 1$, $g = 1$, $h = 1$, **left plot** $\vartheta = 0$ **and right plot** $\vartheta = 0.2$.

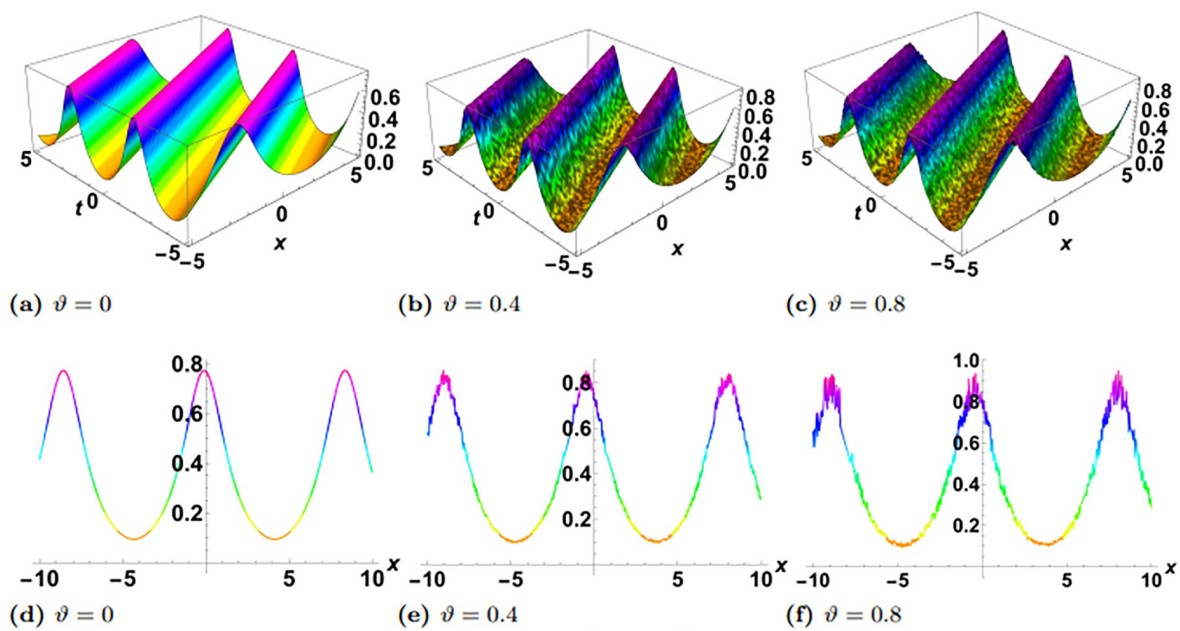

**Fig 9. Visual illustration of the solution $\mathbb{G}_{16}(x, y, z, t)$, taking into account specific parameters** $y = z = 1, \alpha_1 = \alpha_2 = \alpha_3 = 1, \gamma_1 = \gamma_2 = \gamma_3 = \gamma_4 = 1, \xi_0 = 0, \xi_2 = 1, g = 1, h = 1.$

could correspond to stable tidal or surface wave formations. The incorporation of Brownian motion provides a valuable framework to understand how natural noise (e.g., wind or current variations) may affect wave propagation, shape, and stability. Such insights can contribute to improving predictive models for environmental systems or enhancing energy transfer efficiencies in plasma applications.

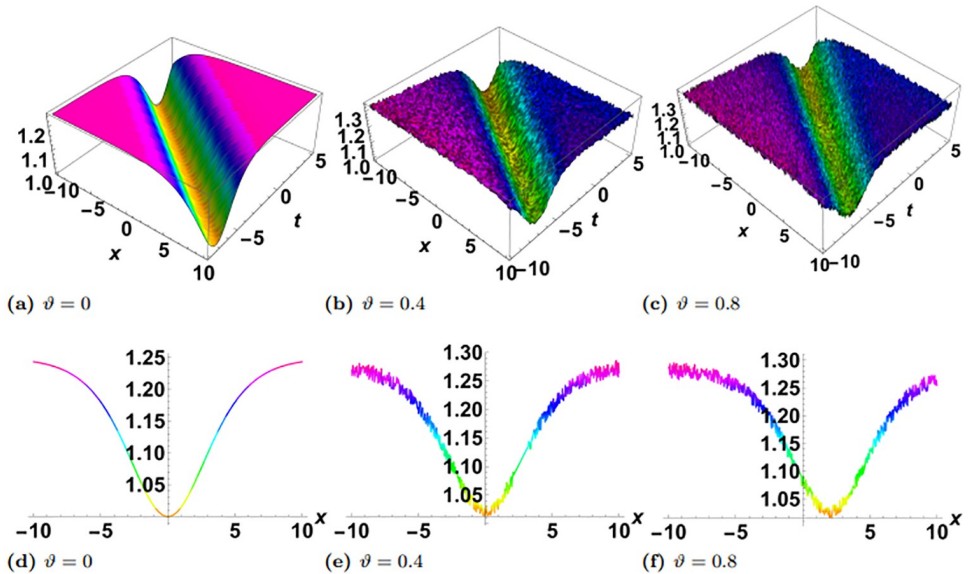

**Fig 10. Visual illustration of the solution $\mathbb{G}_{17}(x, y, z, t)$, taking into account specific parameters** $y = z = 1, \alpha_1 = \alpha_2 = \alpha_3 = 1, \gamma_1 = \gamma_2 = \gamma_3 = \gamma_4 = -1, \xi_0 = 0, \xi_2 = 1, g = 1, h = 1.$

### 5.1 Remarks

In our exploration of randomly selected solutions derived from two distinct approaches, we made an intriguing observation: the solutions generated through the modified tanh method exhibit a predominantly singular nature, with a sprinkle of hybrid ones among them. On the other hand, solutions derived through the SSe approach tend to be mostly non-singular in nature. This diversity in solution characteristics adds a fascinating layer to our understanding of the interplay between different solution methods and their outcomes.

### 5.2 Limitations

The proposed methods have proven effective in generating robust solutions for extended Kairat-II equation, particularly in low to moderate noise environments, they exhibit considerable limitations when encountering high-noise scenarios. In such conditions, the methods can struggle to accurately capture the complex dynamics influenced by strong stochastic perturbations, probably leading to less reliable or physically unrealistic solutions. This limitation is particularly evident in cases where the noise intensity significantly alters the wave structure, resulting in solutions that may not adequately reflect the true behavior of the system.

## 6 Conclusion

In this manuscript, a stochastic version of the extended Kairat-II equation by incorporating Brownian motion in the Ito sense is studied. Through the traveling wave transformation, the suggested equation is transformed into an ODE. Then, two powerful analytical techniques, namely the modified tanh method associated with the Riccati equation and SSEM, have been used to achieve various families of closed-form solutions, which are functions of trigonometric and hyperbolic trigonometric functions. The obtained results are demonstrated via 3D and 2D graphs. The results show distinct wave structures such as bright, dark, singular, and periodic solitons. The effect of Brownian motion is illustrated by varying the noise strength. The findings of this study are expected to impact the modeling of stochastic processes in various physical systems, particularly in areas requiring precise control of wave dynamics under random perturbations. Future research could focus on extending the current model to some more complex stochastic environments or exploring alternative noise-induced models. Additionally, investigating the application of these methods to other integrable systems could further validate the findings. Nowadays, neural network and fractional calculus has several applications in various fields of science [34–36]. So, these approaches can also be employed for important and novel dynamics. Furthermore, the desired results can be applied in plasma physics, optical communication, and marine environments. Additionally, it would be of great interest to study real-world applications, such as improving the design of communication systems or predicting environmental wave patterns.

## Author Contributions

**Conceptualization:** Khaled Aldwoah.

**Formal analysis:** Alaa Mustafa.

**Investigation:** Khidir Mohamed.

**Methodology:** Tariq Aljaaidi.

**Resources:** Mohammed Hassan.

**Software:** Khidir Mohamed, Mohammed Hassan.

**Supervision:** Alaa Mustafa, Mohammed Hassan.

**Validation:** Khaled Aldwoah, Amer Alsulami.

**Writing – original draft:** Amer Alsulami.

**Writing – review & editing:** Alaa Mustafa, Tariq Aljaaidi.

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
