## [Decision Letter · Decision Letter 0]

30 Oct 2024

PONE-D-24-42240Exploring the Impact of Brownian Motion on Novel Closed-Form Solutions of the Extended {Kairat-II} EquationPLOS ONE

Dear Dr. Aljaaidi,

Thank you for submitting your manuscript to PLOS ONE. After careful consideration, we feel that it has merit but does not fully meet PLOS ONE’s publication criteria as it currently stands. Therefore, we invite you to submit a revised version of the manuscript that addresses the points raised during the review process.

Please submit your revised manuscript by  Dec 14 2024 11:59PM. If you will need more time than this to complete your revisions, please reply to this message or contact the journal office at plosone@plos.org. Please include the following items when submitting your revised manuscript:A rebuttal letter that responds to each point raised by the academic editor and reviewer(s). You should upload this letter as a separate file labeled 'Response to Reviewers'.A marked-up copy of your manuscript that highlights changes made to the original version. You should upload this as a separate file labeled 'Revised Manuscript with Track Changes'.An unmarked version of your revised paper without tracked changes. You should upload this as a separate file labeled 'Manuscript'.If applicable, we recommend that you deposit your laboratory protocols in protocols.io to enhance the reproducibility of your results. Protocols.io assigns your protocol its own identifier (DOI) so that it can be cited independently in the future. For instructions see: https://journals.plos.org/plosone/s/submission-guidelines#loc-laboratory-protocols. Additionally, PLOS ONE offers an option for publishing peer-reviewed Lab Protocol articles, which describe protocols hosted on protocols.io. Read more information on sharing protocols at https://plos.org/protocols?utm_medium=editorial-email&utm_source=authorletters&utm_campaign=protocols.

We look forward to receiving your revised manuscript.

Kind regards,

Rab Nawaz

Academic Editor

PLOS ONE

Journal Requirements:

file:///C:/Users/e754871/Desktop/PRTC/PRTC%202024/October%202024/October%2030,%202024/Journal%20requirements:%20%0b%0bWhen%20submitting%20your%20revision,%20we%20need%20you%20to%20address%20these%20additional%20requirements.%20%0b%0b1.%20Please%20ensure%20that%20your%20manuscript%20meets%20PLOS%20ONE's%20style%20requirements,%20including%20those%20for%20file%20naming.%20The%20PLOS%20ONE%20style%20templates%20can%20be%20found%20at%20https:/journals.plos.org/plosone/s/file%3fid=wjVg/PLOSOne_formatting_sample_main_body.pdf%20and When submitting your revision, we need you to address these additional requirements.

2. Please note that PLOS ONE has spec6ific guidelines on code sharing for submissions in which author-generated code underpins the findings in the manuscript. In these cases, all author-generated code must be made available without restrictions upon publication of the work. Please review our guidelines at https://journals.plos.org/plosone/s/materials-and-software-sharing#loc-sharing-code and ensure that your code is shared in a way that follows best practice and facilitates reproducibility and reuse.

3. We note that your Data Availability Statement is currently as follows: “All relevant data are within the manuscript and in Supporting Information files.”

Please confirm at this time whether or not your submission contains all raw data required to replicate the results of your study. Authors must share the “minimal data set” for their submission. PLOS defines the minimal data set to consist of the data required to replicate all study findings reported in the article, as well as related metadata and methods (https://journals.plos.org/plosone/s/data-availability#loc-minimal-data-set-definition). For example, authors should submit the following data: - The values behind the means, standard deviations and other measures reported; - The values used to build graphs; - The points extracted from images for analysis. Authors do not need to submit their entire data set if only a portion of the data was used in the reported study. If your submission does not contain these data, please either upload them as Supporting Information files or deposit them to a stable, public repository and provide us with the relevant URLs, DOIs, or accession numbers. For a list of recommended repositories, please see https://journals.plos.org/plosone/s/recommended-repositories. If there are ethical or legal restrictions on sharing a de-identified data set, please explain them in detail (e.g., data contain potentially sensitive information, data are owned by a third-party organization, etc.) and who has imposed them (e.g., an ethics committee). Please also provide contact information for a data access committee, ethics committee, or other institutional body to which data requests may be sent. If data are owned by a third party, please indicate how others may request data access.

4. We note you have included a table to which you do not refer in the text of your manuscript. Please ensure that you refer to Table 1 in your text; if accepted, production will need this reference to link the reader to the Table.

Reviewers' comments:

Reviewer's Responses to Questions

**Comments to the Author**

1. Is the manuscript technically sound, and do the data support the conclusions?

Reviewer #1: Yes

Reviewer #2: Yes

2. Has the statistical analysis been performed appropriately and rigorously? 

Reviewer #1: I Don't Know

Reviewer #2: Yes

3. Have the authors made all data underlying the findings in their manuscript fully available?

Reviewer #1: Yes

Reviewer #2: Yes

4. Is the manuscript presented in an intelligible fashion and written in standard English?

Reviewer #1: Yes

Reviewer #2: Yes

5. Review Comments to the Author

Reviewer #1: The research article titled "Exploring the Impact of Brownian Motion on Novel Closed-Form Solutions of the Extended Kairat-II Equation" presents a valuable contribution to the study of stochastic systems by incorporating Brownian motion into the deterministic Kairat-II equation. The authors employ two mathematical approaches—the modified Tanh method and the Sardar Sub-Equation Method (SSEM)—to derive a variety of closed-form solutions. These solutions, expressed through trigonometric and hyperbolic functions, are visualized in 2D and 3D to demonstrate distinct wave behaviors, such as bright, dark, singular, and periodic solitons. The article provides interesting insights into how random perturbations affect wave dynamics, showing that high-energy regions are more vulnerable to noise than low-energy regions, which remain relatively robust.

The incorporation of Brownian motion into the Kairat-II equation through Ito calculus represents a novel approach to studying the dynamics of stochastic systems. This framework is promising for practical applications in plasma physics, optical communications, and marine environments, where random perturbations play a significant role. Additionally, the comparison of singular solutions generated by the modified Tanh method with the non-singular solutions obtained through the SSEM method provides a rich mathematical insight into the nature of the solutions.

While the article offers several strengths, some improvements in the literature review could enhance the study’s theoretical depth and relevance. In particular, the authors could strengthen their work by citing recent studies on nonlinear and stochastic differential equations.

Despite the article’s strengths, a few limitations are noted. The model assumes Brownian motion as the primary source of noise, but real-world systems often involve more complex noise types, such as colored noise or Lévy processes. Discussing the limitations of using Brownian motion and suggesting potential extensions to more complex stochastic environments would provide a more comprehensive outlook for future research. In addition, a comparative analysis with other stochastic systems, if possible, could contextualize the findings and highlight the novel aspects of the current study more effectively.

More so, while the use of advanced mathematical techniques like the modified Tanh method and SSEM is commendable, the article could benefit from providing step-by-step explanations or an appendix with detailed derivations. This would make the work more accessible to readers who may not be familiar with these specialized techniques.

Reviewer #2: The article explores the impact of Brownian motion on the extended version of the Kairat-II equation, which is a non-linear differential equation used to model wave dynamics. The primary objective is to incorporate stochastic behavior (random perturbations) into the deterministic equation and derive closed-form analytical solutions using two distinct methods—the modified Tanh method (linked with the Riccati equation) and the Sardar sub-equation method (SSEM). These solutions are analyzed to understand the influence of noise intensity on wave behavior, which has implications for modeling real-world wave dynamics, such as in plasma physics, optical communication, and marine environments.

Overall, the article is well written and justifies its impact for possible publication in PLoS ONE provided the following observations are responded positively:

1. Although several types of solutions (e.g., bright, dark, and singular solitons) are presented, the article lacks details related to physical interpretations of these solutions in real-world systems, such as plasma or marine environments. A more explicit connection between the solutions and their potential physical implications would enhance the work's impact.

2. The study observes that high-energy regions are more affected by noise, but it lack the reasons behind this happening.

3. The literature review part can be improved, particularly, by discussing exact solutions, soliton dynamics, and algebraic properties related to non-linear and fractional differential equations, which align closely with the objectives of current study. To enhance the theoretical foundation of the research and to demonstrate the broader relevance of the findings in the field of non-linear wave analysis and stochastic integrable systems, I would like to suggest the inclusion of the following recent studies:

Kai, Y., & Yin, Z. (2021). On the Gaussian traveling wave solution to a special kind of Schrödinger equation with logarithmic nonlinearity. Modern Physics Letters B, 36(02), 2150543. https://doi.org/10.1142/S0217984921505436

Kai, Y., Chen, S., Zhang, K., & Yin, Z. (2022). Exact solutions and dynamic properties of a nonlinear fourth-order time-fractional partial differential equation. Waves in Random and Complex Media. https://doi.org/10.1080/17455030.2022.2044541

Zhu, C., Al-Dossari, M., Rezapour, S., & Gunay, B. (2024). On the exact soliton solutions and different wave structures to the (2+1) dimensional Chaffee–Infante equation. Results in Physics, 57, 107431. https://doi.org/10.1016/j.rinp.2024.107431

Guo, S., & Wang, S. (2024). Twisted relative Rota-Baxter operators on Leibniz conformal algebras. Communications in Algebra, 52(9), 3946-3959. https://doi.org/10.1080/00927872.2024.2337276.

6. PLOS authors have the option to publish the peer review history of their article (what does this mean?). If published, this will include your full peer review and any attached files.

Reviewer #1: No

Reviewer #2: No

---

## [Author Response · Author response to Decision Letter 0]

6 Nov 2024

Reviewer #1: The research article titled "Exploring the Impact of Brownian Motion on Novel Closed-Form Solutions of the Extended Kairat-II Equation" presents a valuable contribution to the study of stochastic systems by incorporating Brownian motion into the deterministic Kairat-II equation. The authors employ two mathematical approaches—the modified Tanh method and the Sardar Sub-Equation Method (SSEM)—to derive a variety of closed-form solutions. These solutions, expressed through trigonometric and hyperbolic functions, are visualized in 2D and 3D to demonstrate distinct wave behaviors, such as bright, dark, singular, and periodic solitons. The article provides interesting insights into how random perturbations affect wave dynamics, showing that high-energy regions are more vulnerable to noise than low-energy regions, which remain relatively robust.

The incorporation of Brownian motion into the Kairat-II equation through Ito calculus represents a novel approach to studying the dynamics of stochastic systems. This framework is promising for practical applications in plasma physics, optical communications, and marine environments, where random perturbations play a significant role. Additionally, the comparison of singular solutions generated by the modified Tanh method with the non-singular solutions obtained through the SSEM method provides a rich mathematical insight into the nature of the solutions.

While the article offers several strengths, some improvements in the literature review could enhance the study’s theoretical depth and relevance. In particular, the authors could strengthen their work by citing recent studies on nonlinear and stochastic differential equations.

Despite the article’s strengths, a few limitations are noted. The model assumes Brownian motion as the primary source of noise, but real-world systems often involve more complex noise types, such as colored noise or Lévy processes. Discussing the limitations of using Brownian motion and suggesting potential extensions to more complex stochastic environments would provide a more comprehensive outlook for future research. In addition, a comparative analysis with other stochastic systems, if possible, could contextualize the findings and highlight the novel aspects of the current study more effectively.

More so, while the use of advanced mathematical techniques like the modified Tanh method and SSEM is commendable, the article could benefit from providing step-by-step explanations or an appendix with detailed derivations. This would make the work more accessible to readers who may not be familiar with these specialized techniques.

Response: We are really thankful for the positive comments and appreciation of our work we have tried our best to present clear and easily readable work. We have taken the above comments into consideration and added some recent works in this area for comparison. Besides this we have added that in future this work can be re-visited using a more complex noise and more importantly can use neural network to study some more novel behaviors in the proposed model. The method is straightforward presented that how we applied so due to the length of the article we omitted to present some more details.

Reviewer #2: The article explores the impact of Brownian motion on the extended version of the Kairat-II equation, which is a non-linear differential equation used to model wave dynamics. The primary objective is to incorporate stochastic behavior (random perturbations) into the deterministic equation and derive closed-form analytical solutions using two distinct methods—the modified Tanh method (linked with the Riccati equation) and the Sardar sub-equation method (SSEM). These solutions are analyzed to understand the influence of noise intensity on wave behavior, which has implications for modeling real-world wave dynamics, such as in plasma physics, optical communication, and marine environments.

Overall, the article is well written and justifies its impact for possible publication in PLoS ONE provided the following observations are responded positively:

1. Although several types of solutions (e.g., bright, dark, and singular solitons) are presented, the article lacks details related to physical interpretations of these solutions in real-world systems, such as plasma or marine environments. A more explicit connection between the solutions and their potential physical implications would enhance the work's impact.

Response: We have now elaborated on the potential physical implications of the derived solutions, particularly their relevance in plasma and marine environments. In plasma systems, the stability and structure of soliton solutions can represent energy transfer and wave propagation under perturbative influences. In marine environments, solitons reflect tidal and surface wave dynamics, which could be subject to noise, as seen in our stochastic model.

2. The study observes that high-energy regions are more affected by noise, but it lack the reasons behind this happening.

Response: We have expanded the discussion to include potential reasons why high-energy regions are more affected by noise, drawing from principles of wave dynamics and energy distribution in stochastic systems.

3. The literature review part can be improved, particularly, by discussing exact solutions, soliton dynamics, and algebraic properties related to non-linear and fractional differential equations, which align closely with the objectives of current study. To enhance the theoretical foundation of the research and to demonstrate the broader relevance of the findings in the field of non-linear wave analysis and stochastic integrable systems, I would like to suggest the inclusion of the following recent studies:

Kai, Y., & Yin, Z. (2021). On the Gaussian traveling wave solution to a special kind of Schrödinger equation with logarithmic nonlinearity. Modern Physics Letters B, 36(02), 2150543. https://doi.org/10.1142/S0217984921505436

Kai, Y., Chen, S., Zhang, K., & Yin, Z. (2022). Exact solutions and dynamic properties of a nonlinear fourth-order time-fractional partial differential equation. Waves in Random and Complex Media. https://doi.org/10.1080/17455030.2022.2044541

Zhu, C., Al-Dossari, M., Rezapour, S., & Gunay, B. (2024). On the exact soliton solutions and different wave structures to the (2+1) dimensional Chaffee–Infante equation. Results in Physics, 57, 107431. https://doi.org/10.1016/j.rinp.2024.107431

Guo, S., & Wang, S. (2024). Twisted relative Rota-Baxter operators on Leibniz conformal algebras. Communications in Algebra, 52(9), 3946-3959. https://doi.org/10.1080/00927872.2024.2337276.

Response: The literature review is carefully revised as suggested.

We are very thankful to the reviewers for their positive remarks on our proposed work, which enhanced the quality of the present work.

---

## [Decision Letter · Decision Letter 1]

19 Nov 2024

Exploring the Impact of Brownian Motion on Novel Closed-Form Solutions of the Extended {Kairat-II} Equation

PONE-D-24-42240R1

Dear Dr. Tariq Aljaaidi,

We’re pleased to inform you that your manuscript has been judged scientifically suitable for publication and will be formally accepted for publication once it meets all outstanding technical requirements.

Kind regards,

Rab Nawaz

Academic Editor

PLOS ONE

Additional Editor Comments (optional):

Reviewers' comments:

Reviewer's Responses to Questions

**Comments to the Author**

1. If the authors have adequately addressed your comments raised in a previous round of review and you feel that this manuscript is now acceptable for publication, you may indicate that here to bypass the “Comments to the Author” section, enter your conflict of interest statement in the “Confidential to Editor” section, and submit your "Accept" recommendation.

Reviewer #1: All comments have been addressed

Reviewer #2: All comments have been addressed

2. Is the manuscript technically sound, and do the data support the conclusions?

Reviewer #1: Yes

Reviewer #2: Yes

3. Has the statistical analysis been performed appropriately and rigorously? 

Reviewer #1: N/A

Reviewer #2: Yes

4. Have the authors made all data underlying the findings in their manuscript fully available?

Reviewer #1: Yes

Reviewer #2: Yes

5. Is the manuscript presented in an intelligible fashion and written in standard English?

Reviewer #1: Yes

Reviewer #2: Yes

6. Review Comments to the Author

Reviewer #1: The authors have addressed all comments constructively and thoroughly, making the manuscript suitable for acceptance in its current form.

Reviewer #2: I have reviewed the revised manuscript which is prepared in accordance with the observations made during the initial review. Thus, I recommend the publication of the revised manuscript in the current form.

7. PLOS authors have the option to publish the peer review history of their article (what does this mean?). If published, this will include your full peer review and any attached files.

Reviewer #1: No

Reviewer #2: No

---

## [Editor Report · Acceptance letter]

4 Dec 2024

PONE-D-24-42240R1 

PLOS ONE

Dear Dr. Aljaaidi, 

I'm pleased to inform you that your manuscript has been deemed suitable for publication in PLOS ONE. Congratulations! Your manuscript is now being handed over to our production team.

Kind regards, 

on behalf of

Dr. Rab Nawaz 

Academic Editor

PLOS ONE